# Practicality and Robustness of Tree Species Identification Using UAV RGB Image and Deep Learning in Temperate Forest in Japan

**Masanori Onishi [1,\*], Shuntaro Watanabe [2], Tadashi Nakashima [3] and Takeshi Ise [3]**

[1] Graduate School of Agriculture, Kyoto University, Kyoto 606-8502, Japan
[2] Graduate School of Science and Engineering, Kagoshima University, Kagoshima 890-0065, Japan; watanabe@sci.kagoshima-u.ac.jp
[3] Field Science Education and Research Center, Kyoto University, Kyoto 606-8502, Japan; nakashima.tadashi.p83@kyoto-u.jp (T.N.); ise.takeshi.5e@kyoto-u.ac.jp (T.I.)
[\*] Correspondence: onishi.masanori.25e@kyoto-u.jp

**Abstract:** Identifying tree species from the air has long been desired for forest management. Recently, combination of UAV RGB image and deep learning has shown high performance for tree identification in limited conditions. In this study, we evaluated the practicality and robustness of the tree identification system using UAVs and deep learning. We sampled training and test data from three sites in temperate forests in Japan. The objective tree species ranged across 56 species, including dead trees and gaps. When we evaluated the model performance on the dataset obtained from the same time and same tree crowns as the training dataset, it yielded a Kappa score of 0.97, and 0.72, respectively, for the performance on the dataset obtained from the same time but with different tree crowns. When we evaluated the dataset obtained from different times and sites from the training dataset, which is the same condition as the practical one, the Kappa scores decreased to 0.47. Though coniferous trees and representative species of stands showed a certain stable performance regarding identification, some misclassifications occurred between: (1) trees that belong to phylogenetically close species, (2) tree species with similar leaf shapes, and (3) tree species that prefer the same environment. Furthermore, tree types such as coniferous and broadleaved or evergreen and deciduous do not always guarantee common features between the different trees belonging to the tree type. Our findings promote the practicalization of identification systems using UAV RGB images and deep learning.

**Keywords:** deep learning; UAV RGB image; tree species identification

## 1. Introduction

Identifying tree species from data obtained from remote sensing has been desired for forest management processes such as management and protection of native vegetation [1], monitoring invasive species [2], and wild habitat mapping [3]. In Japan, tree species identification and mapping at large scale have been required for effective logging and sustainable forest management. In forestry, much of the planted forest has been abandoned due to the low timber price. To use these forests effectively, surveys of forest resources including species information have been conducted on a large scale. In addition, to sustainably fulfill the multifunctional role of forests in preventing mountain disasters and conserving soil, some planted forests are being converted to mixed needle and broadleaf forests, and broadleaf forests [4]. For these purposes, the ability to obtain information on tree species over a wide area makes it possible to easily zone areas that need to be treated and to monitor their progress.

For those applications, studies using multispectral sensors have been conducted since before the 1980s, but the number of studies using hyperspectral sensors from aircrafts has

been increasing since around 2010, with higher accuracy than multispectral sensors [5–7]. Although a general digital camera can capture the spectra of three bands of red, green, and blue within visible areas, multispectral cameras can capture 4 to 8 bands (400–700 nm), including near infrared (NIR: 700–1300 nm), and hyperspectral cameras can catch from 100 to 200 bands, including NIR and short-wave infrared (SWIR: 1300–2500 nm). The reason for using these invisible spectra is for identification, because various spectra reflecting tree crowns have a relationship with (1) chemical properties such as water in the woody tissue, photosynthetic pigment, and characteristics of structural carbohydrates [8–11], (2) morphology of leaves [8,12,13]. To apply machine learning, hundreds or thousands of spectra sensed using hyperspectral sensors were sometimes compressed dimensionally by PCA, and by applying machine learning such as random forest or SVM, several tree species can be identified with a high accuracy of approximately 80–90% [6,7]. Another approach is LiDAR, which is used for tree detection and also tree species identification. Even by itself, certain characteristic tree species, such as *Cryptomeria japonica* and *Chamaecyparis obtuse*, are detected using features such as reflectance strength and crown shapes [14]. The combination of hyperspectral sensors and LiDAR showed the highest accuracy for the identification of several species [6,7]. In urban areas, the combination also showed high potential not only for identification of several of each tree species but also for estimating tree carbon storages or tree damages because the sensors can catch various chemical tree properties [15–17]. The advantage of using airborne sensors is that they can scan large areas, such as prefecture scales; however, they are costly, and the system is easily affected by several factors. It is expensive; therefore, only huge projects or administrations can apply the method on rare occasions.

For practical use, the temporal and spatial robustness should be considered. However, these studies were conducted using the data obtained simultaneously, and most were obtained from one site [2,6,7,9,18–24]. The accuracy could drop if identification is performed in a different region or at a different time of year than the training data because the spectra can be affected by shadows and illumination owing to the weather, background signals attributed to the conditions of vegetation, soil and litter, density of leaves, and health of trees [2,6,7,9,19,20,25].

Recently, UAVs have been used as a framework for tree identification. Some studies have loaded hyperspectral sensors for identifying trees [26,27]. However, other studies have identified tree species from digital cameras helped by deep learning [28–30], because of the high resolution of images and deep learning, specific tree species detection, classification, and mapping of several tree species or tree types. The advantages of this system are its low cost and the potential for robustness. Onishi and Ise [29] successfully identified several tree classes using Phantom 4 (DJI), which costs less than 3000 USD; because of their low cost, the UAV is used frequently. The system may have the potential for robustness. This means that the model trained in one place has the potential to be used on another day or place, because this system does not rely on the spectral reflectance information. With the help of high-resolution images and deep learning, trees are identified using the texture images generated by leaf shapes and branching patterns [29]. The texture appearance may also be affected by some conditions such as wind strength, flight and camera parameters (e.g., ground sampling distance (GSD), flight speed, shutter speed), and the image processing for compositing an entire image from images with low GPS. However, under similar conditions, the model is expected to perform well, even if the location or timing of the data differs from that of the training dataset. The disadvantage is that UAVs can only cover limited areas compared to the airplane. Multi-copter UAVs such as phantoms can fly for less than 40 min, and only approximately 10 ha can be scanned in one flight.

For the practical use of tree species identification from UAVs, we should explore their potential and robustness. Potential refers to the limit of the amount or kinds of tree species that can be identified. Although the UAV and deep learning can identify trees based on texture information, the texture may not be different for every tree species. In addition, the spatial and temporal robustness of the model should be assessed for general usage.

In this study, we used general UAVs and deep learning, collected large amounts of tree species data from temperate forests in Japan, and tested the robustness using the datasets obtained: (1) at the same acquisition date or from same individual tree, (2) at the same acquisition date and from different individual trees, and (3) at different acquisition date and at different sites.

## 2. Materials and Methods

### 2.1. Study Sites

This study was conducted at six sites in a temperate forest in Japan. We used three sites for training and validation, and three sites for testing; the sites are outlined below (Figure 1, Table 1). The study sites were chosen because they contained matured cold-temperate forest, warm-temperate forest, and major species of planted forest such as *Chamaecyparis obtuse* and *Cryptomeria japonica* in both training and test sites.

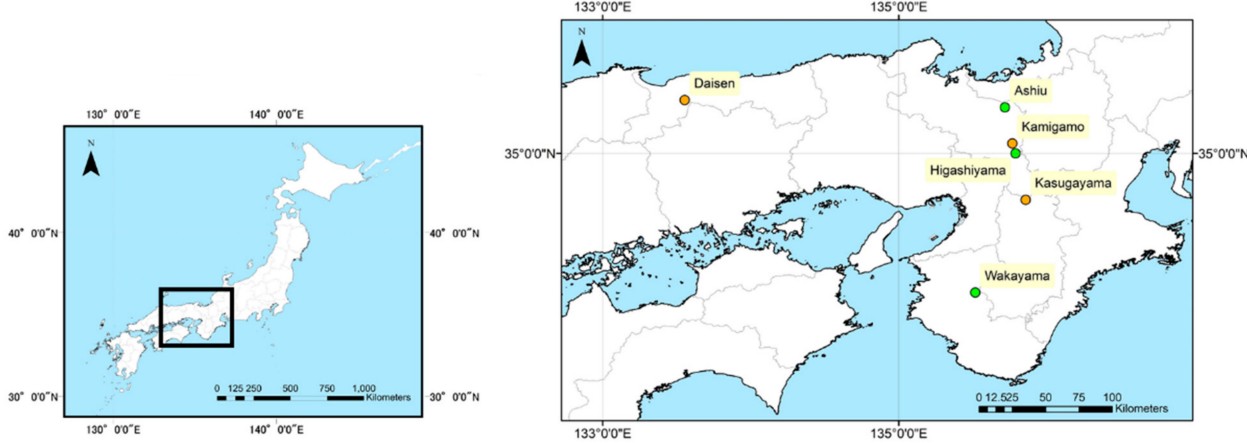

**Figure 1.** Location of each site. Green and orange points represent the training and test sites, respectively.

**Table 1.** Site information summary.

| Site Name | Higashiyama | Wakayama | Ashiu |
| --- | --- | --- | --- |
| Data class | Training | Training | Training |
| Field research area | 2 ha | 1 ha | 4 ha |
| Flight date | 4, 5 July and 14, 15 September 2019 | 15, 17 July and 8 October 2019 | 24–26 July and 18–20 September 2019 |
| dominant species | *Castanopsis cuspidate* *Quercus serrata* | *Abies firma*, *Tsuga sieboldii* | *Fagus crenata* *Cryptomeria japonica* |
| **Site Name** | **Kamigamo** | **Kasugayama** | **Daisen** |
| Data class | Test | Test | Test |
| Field research area | 1.5 ha | 1 ha | 1 ha |
| Flight date | 10 October 2019 | 24 September 2019 | 1 October 2019 |
| dominant species | *Chamaecyparis obtusa* *Quercus serrata* | *Castanopsis cuspidate* *Abies firma* | *Fagus crenata* |

- Higashiyama site: located in (34°59′58″N, 135°47′17″E), it is a secondary forest of warm-temperate forest, dominated by deciduous broadleaved trees such as *Quercus serrata*, and laurel evergreen broadleaved trees *such as Castanopsis cuspidate*.
- Wakayama site: located in Wakayama Forest Research Station of Kyoto University (34°03′47″N, 135°31′00″E), it is a natural forest of mid-temperate forest dominated by evergreen coniferous trees such as *Abies firma* and *Tsuga sieboldii*. We used field survey data of Monitoring Sites 1000 Project [31].

- Ashiu site: located at Yusen Valley in Ashiu Forest Research Station of Kyoto University (35°18′34″N, 135°43′1″E), it is a natural forest of cold-temperate forest dominated by deciduous broadleaved trees such as *Fagus crenata* and *Cryptomeria japonica*.
- Kamigamo site: located at Kamigamo Experimental Station of Kyoto University (35°04′00″N, 135°46′01″E), it is a natural regeneration forest of warm-temperate forests dominated by evergreen coniferous trees such as *Chamaecyparis obtuse* and broadleaved trees such as *Quercus serrata*.
- Kasugayama site: located in the Kasugayama Primeval Forest (34°41′14″N, 135°51′24″E), it is a primeval forest of mid-temperate forest dominated by evergreen coniferous trees such as *Abies firma* and laurel evergreen broadleaved trees such as *Castanopsis cuspidate*.
- Daisen site: located in (35°21′32″N, 133°33′17″E), it is a naturally generated forest of cold-temperate forest dominated by deciduous broadleaved trees such as *Fagus crenata*. We used field survey data of Monitoring Sites 1000 Project [31].

### 2.2. UAV Flight

We flew a UAV in the summer of 2019. At the training site, we flew the UAV several times to obtain the training data that looked different owing to the influence of light and weather conditions. At the test site, the UAV was flown once. At both sites, flight timing was set to 1 h before and after noon to avoid shadows, and we mainly used data obtained on a cloudy day. During the flight, we used Phantom 4 pro (DJI, Shenzhen, China) with a 20-million-pixel camera, and UgCS v3.2 software (SPH engineering, Baložu pilsēta, Latvia) for automatic flight. The flight parameters were set as follows: the flight altitude was 100 m from the 10 m resolution DEM, which was provided by the Geospatial Information Authority of Japan. The front overlap was 90%, and the side overlap was 80%. The image format was set as RAW, exposure was set to 0.0, and other camera parameters were set automatically. We did not set GCPs on the ground.

### 2.3. Field Survey

We conducted a field survey in 2019, excluding the Kasugayama site in 2020. In the field survey, we identified every tree species and attached the name as a label to all tree crown images in orthomosaic photos. For label attachment, a specific tree that stood out in the image was mapped to a field and the tree canopy next to it was mapped based on orientation and crown shape. This gave 13,937 tree crown images at the objective area obtained by the UAV, and collected 58 species from the three training sites (Table 2).

**Table 2.** Tree species list of training data which was identified in UAV imagery from the field survey.

| Class | Species Name | Class | Species Name |
|---|---|---|---|
| 1 | *Chamaecyparis obtusa* | 30 | *Quercus serrata* |
| 2 | *Cryptomeria japonica* | 31 | *Pterocarya rhoifolia* |
| 3 | *Abies firma* | 32 | *Cinnamomum camphora* |
| 4 | *Pinus densiflora* | 33 | *Magnolia obovata* |
| 5 | *Tsuga sieboldii* | 34 | *Magnolia salicifolia* |
| 6 | *Ilex chinensis* | 35 | *Morella rubra* |
| 7 | *Ilex latifolia* | 36 | *Fraxinus lanuginosa f. serrata* |
| 8 | *Ilex macropoda* | 37 | *Ternstroemia gymnanthera* |
| 9 | *Ilex micrococca* | 38 | *Hovenia dulcis* |
| 10 | *Ilex pedunculosa* | 39 | *Hovenia tomentella* |
| 11 | *Chengiopanax sciadophylloides* | 40 | *Aria alnifolia* |
| 12 | *Evodiopanax innovans* | 41 | *Aria japonica* |
| 13 | *Kalopanax septemlobus* | 42 | *Malus tschonoskii* |
| 14 | *Betura grossa* | 43 | *Prunus grayana* |
| 15 | *Carpinus cordata* | 44 | *Prunus jamasakura* |
| 16 | *Carpinus japonica* | 45 | *Meliosma Myriantha* |
| 17 | *Carpinus laxiflora* | 46 | *Populus tremula var. sieboldii* |

**Table 2.** *Cont.*

| Class | Species Name | Class | Species Name |
|---|---|---|---|
| 18 | *Carpinus tschonoskii* | 47 | *Acer carpinifolium* |
| 19 | *Ostrya japonica* | 48 | *Acer mono Maxim* |
| 20 | *Cercidiphyllum japonicum* | 49 | *Acer nipponicum* |
| 21 | *Lyonia ovalifolia var.elliptica* | 50 | *Acer palmatum* |
| 22 | *Castanea crenata* | 51 | *Acer palmatum var. amoenum* |
| 23 | *Castanopsis cuspidata* | 52 | *Acer sieboldianum* |
| 24 | *Fagus crenata* | 53 | *Aesculus turbinata* |
| 25 | *Fagus japonica* | 54 | *Symplocos prunifolia* |
| 26 | *Quercus acuta* | 55 | *Stewartia monadelpha* |
| 27 | *Quercus crispula* | 56 | *Zelkova serrata* |
| 28 | *Quercus glauca* | 57 | *dead_tree* |
| 29 | *Quercus salicina* | 58 | *Gap* |

## 2.4. UAV Data Processing

The UAV data processing method is illustrated in Figure 2. First, from UAV imagery, we created an orthomosaic photo and digital surface model (DSM) using Metashape software (Agisoft LLC, St. Petersburg, Russia). From the DSM, we created a slope model, calculated the slope using ArcGIS Desktop v10.6 software (Environmental Systems Research Institute, Inc., Redlands, CA, USA), and orthomosaic photo, DSM, and slope, and applied multiresolution segmentation [32] in eCognition Developer v9.0.0 software (Trimble, Inc., Sunnyvale, CA, USA), and obtained tree crown polygons. Then, we attached species labels to each tree crown polygon using the field survey results, and applied the ExtractByMask tool to extract each tree crown image in ArcGIS. For the training dataset, we took orthomosaic photos from each flight, reduced the GPS error of each image by georeferencing using at least four prominent locations, such as tree tops and artifacts, and used orthomosaic photos for image extraction. From the UAV data processing above, we obtained the supervised data (Figure 3). For the training dataset, the number of images was the same as the number of polygons multiplied by the number of flights.

## 2.5. Deep Learning

We used PyTorch [33] as the deep learning framework, and EfficientNet B7 [34] as the convolutional neural network (CNN) model, and applied fine-tuning: the output layer was changed to 58 classes, and all parameters were trained. Other parameters were set to {batch: 8, learning rate: 0.005, optimizer: SGD, momentum: 0.9, epochs 200}. In the training phase, we augmented the training image number, which was less than 100 to approximately 100, by random rotation and horizontal flip.

We also developed a method for improving classification performance using the vegetation data of the target area, referred to as inventory tuning. Ideally, the model that trained only the tree species that existed in the target area showed optimal classification accuracy. However, making the local model for every target area is not realistic because the vegetation is constantly changing, and hundreds of models must be created. Conversely, in inventory tuning, we use one model trained by many classes (in this case, 58 classes), and in the prediction phase, restrict output to only the tree species listed in the vegetation data by adjusting the output probability results of each class (Figure 4). Using this system, we can create a local model from a big model without retraining. In this study, we made a local model for each test site, respectively, using the tree species information obtained by field survey.

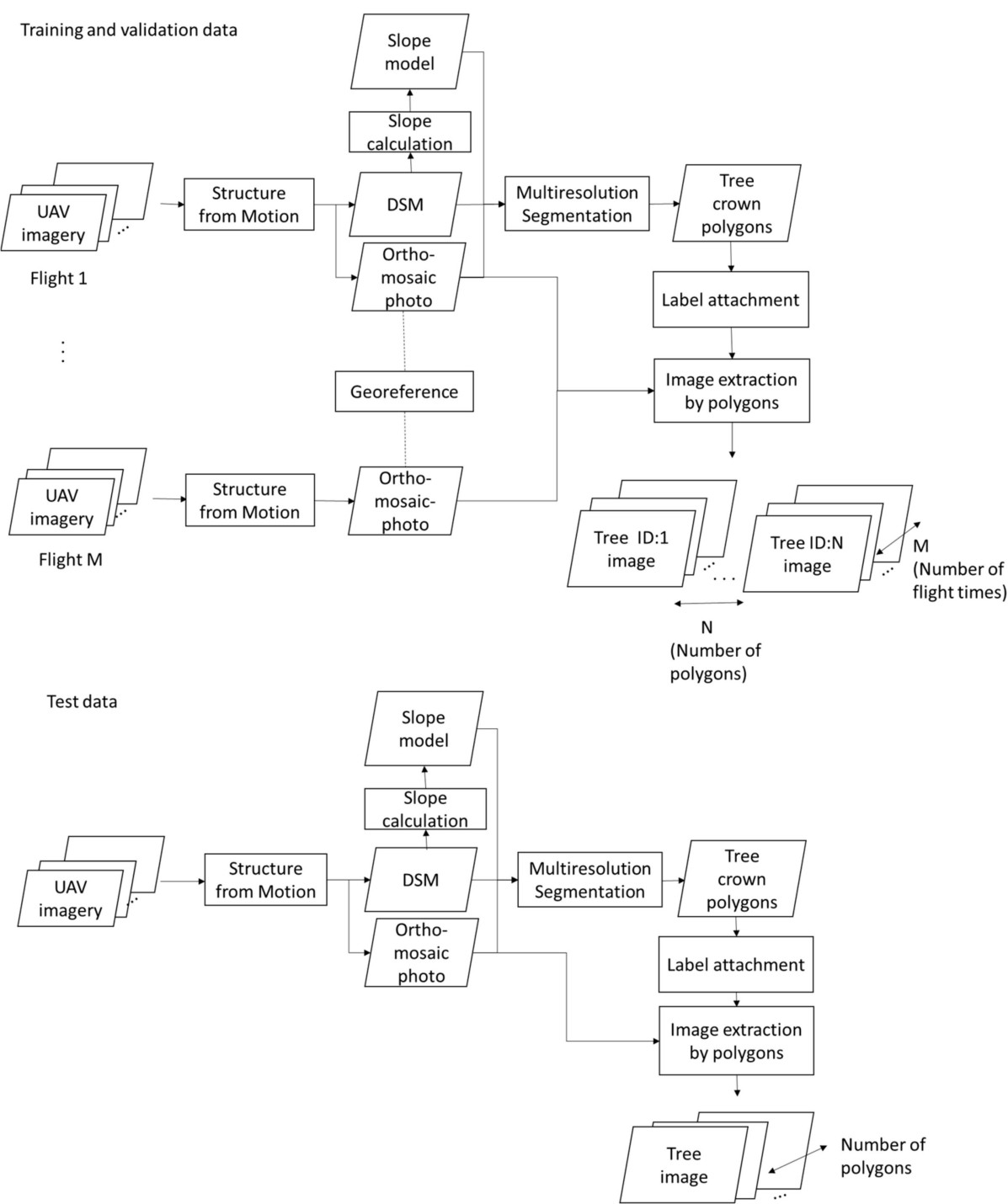

**Figure 2.** UAV imagery processing workflow. We made tree crown polygons using three-dimensional data and an orthomosaic photo, and extracted individual tree crown images. From this process, in the training dataset, the image of the individual tree is obtained for the number of flights, and the amount of training data is obtained by multiplying the number of flights by the number of polygons. In the test dataset, the flight was conducted only once, thus the amount of test data is the same as the number of tree crown polygons.

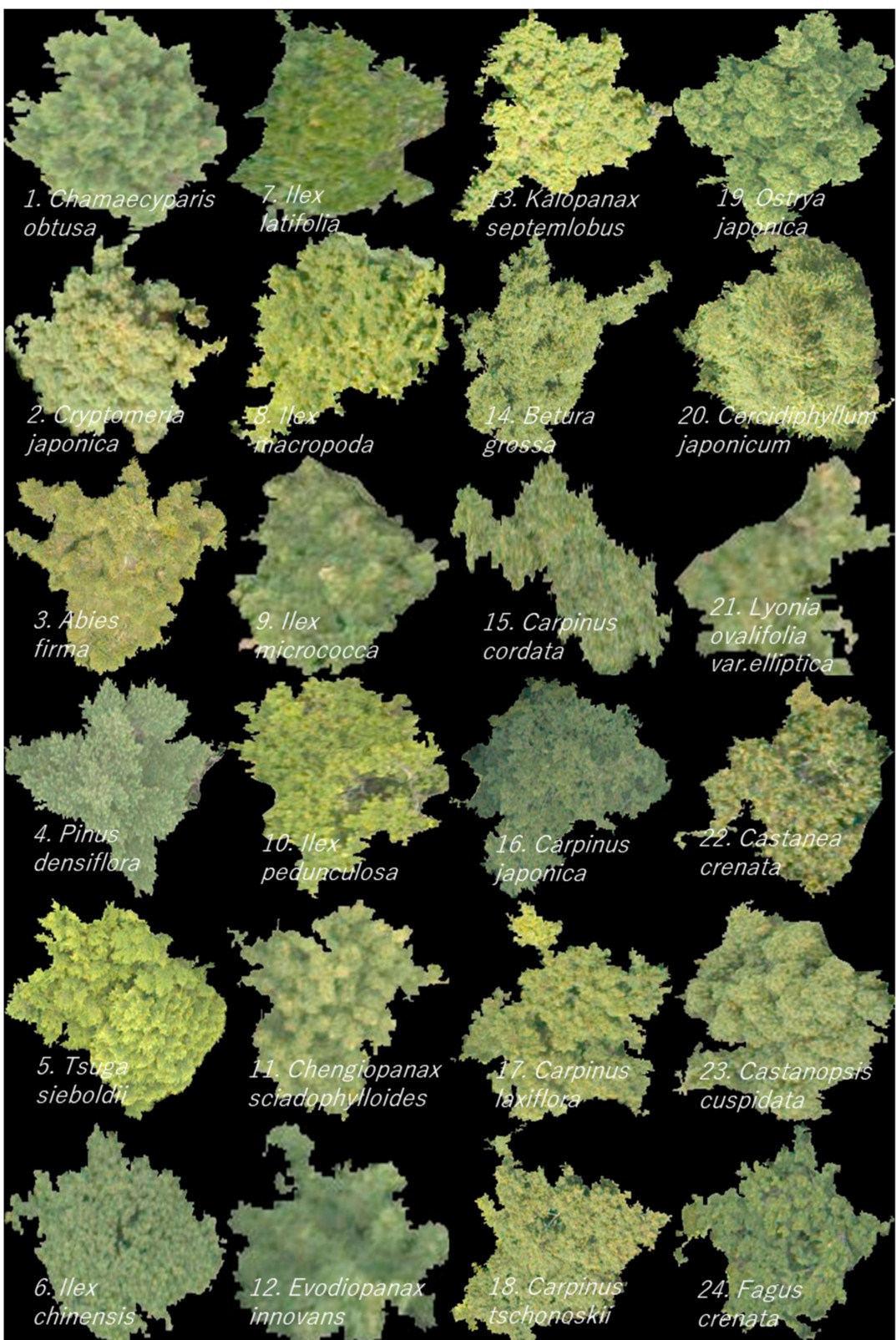

**Figure 3.** *Cont.*

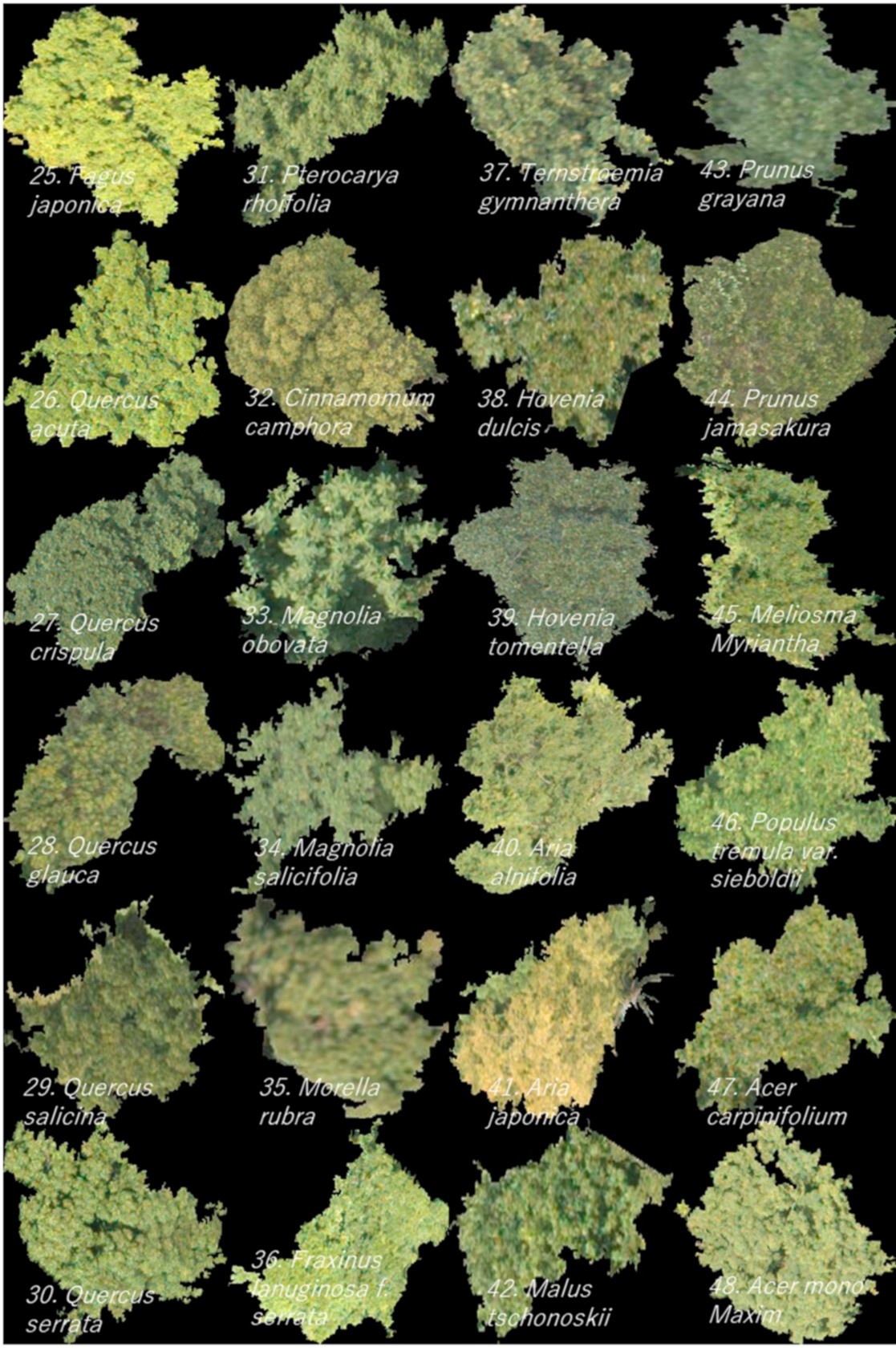

**Figure 3.** *Cont.*

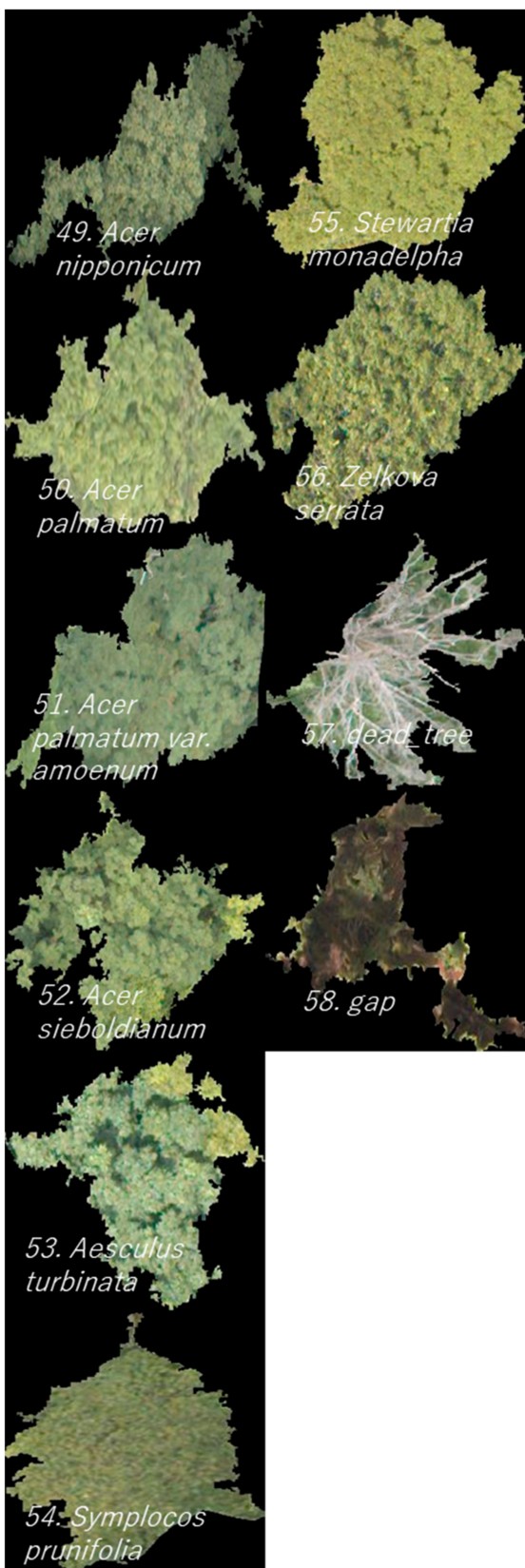

**Figure 3.** Tree crown images of each species taken by UAV from 100 m altitude.

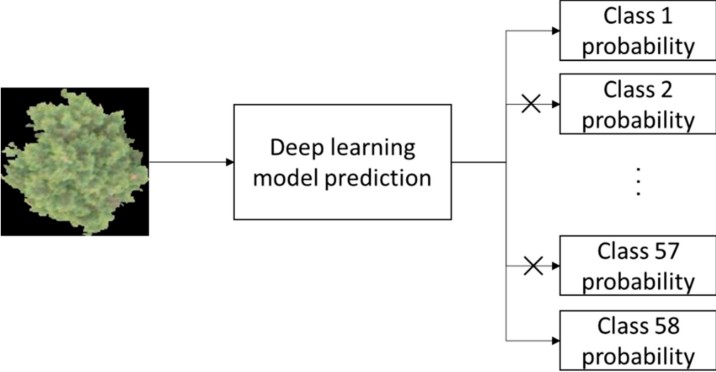

**Figure 4.** Image of inventory tuning.

## 2.6. Analysis

We assessed the accuracy of the two types of validation and one test. The separation method of the validation dataset is shown in Figure 5. In validation 1, we divided the dataset randomly into training and validation datasets at rates of 70% and 30%, respectively. In the validation 1 dataset, images of the same individual tree obtained at various acquisition times can be obtained in both the training and validation dataset. Validation 1 can evaluate the performance of the model for the dataset containing the images which were taken at the same acquisition date but of different individual trees and images of same individual tree but taken at a different acquisition date. In validation 2, the images were divided by polygons at the same rate as validation 1. Therefore, the tree ID:1 images are included only in training, and tree ID:2 images are included in validation. Thus, validation 2 showed the performance of the model at the same acquisition date but with different individual tree crowns. The number of images for each dataset is shown in Supplementary Table S1. The number of images varied widely, with the smallest class having only six images and the largest class having 1624 images. The test data included 26 of the 58 classes.

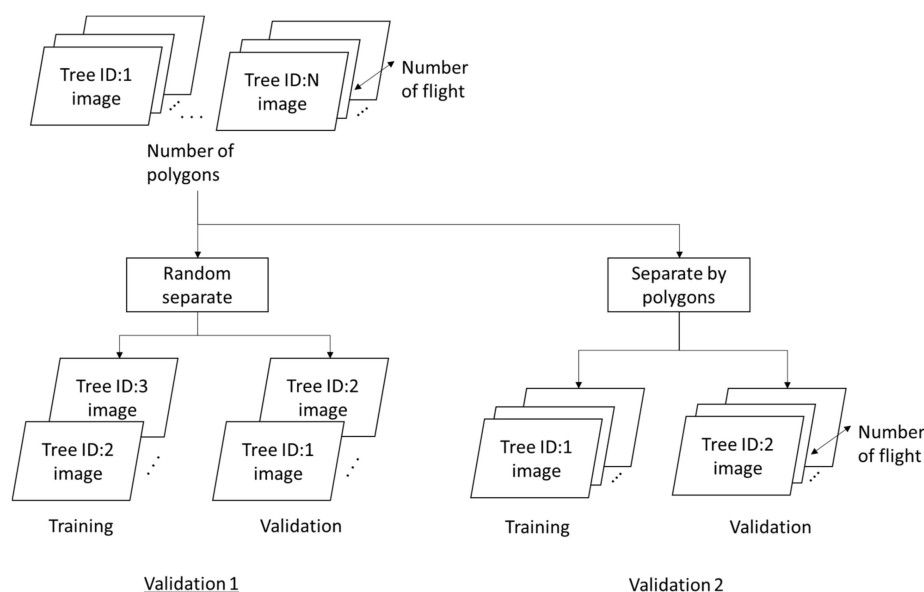

**Figure 5.** Method for validation dataset separation. The images were extracted from a previous process shown in upper process diagram in Figure 2. In validation 1, training and validation datasets were divided randomly. Thus, this dataset contains the images which were taken at same acquisition date but of different individual trees and images of same individual tree but taken at different acquisition times. In validation 2, the training and validation datasets were divided by polygons. Thus, this dataset contains same acquisition date but with different individual tree crowns.

*2.7. Performance Evaluation*

For performance evaluation, we used Cohen's Kappa score [35] to evaluate the overall performance, and precision and recall for each class classification accuracy. To visualize the details of the classification result, we used a confusion matrix where the vertical axis indicates the ground truth label, and the horizontal axis shows the prediction label. The number of each cell represents the number of images. However, this method cannot confirm the misclassification between each class because each cell must be small when the number of classes is large. To visualize the misclassification between classes, which means similarities across classes, we developed a visualization method using a dendrogram.

Here, we considered this matrix (Table 3), and calculated the classification accuracy of class A; the *F*1 score is one of the major accuracy indices. The *F*1 score is the harmonic mean of the precision and recall. Precision, recall, and *F*1 score can be calculated using the formula below.

$$Precision_A = \frac{a}{a + d + g} \tag{1}$$

$$Recall_A = \frac{a}{a + b + c} \tag{2}$$

$$F1\ score = \frac{2 \times Recall \times Precision}{Recall + Precision} \tag{3}$$

**Table 3.** Sample matrix of classification result.

| | | Model Prediction | | |
|---|---|---|---|---|
| | Class | A | B | C |
| Ground truth | A | a | b | c |
| | B | d | e | f |
| | C | g | h | We |

Focusing on the classification accuracy between classes *A* and *C*, we calculated the performance using Equations (4) and (5) below.

$$F1\ score_A\ between\ A\ and\ C = \frac{2a}{2a + c + g} \tag{4}$$

$$F1\ score_C\ between\ A\ and\ C = \frac{2i}{2i + c + g} \tag{5}$$

From these two *F*1 scores, the identification accuracy between *A* and *C* can be calculated by averaging the scores.

$$F1\ score\ between\ A\ and\ C = \left( \frac{2a}{2a + c + g} + \frac{2i}{2i + c + g} \right) \times \frac{1}{2} = \frac{4ai + (a + 1)(c + g)}{(2a + c + g)(2i + c + g)} \tag{6}$$

A low *F*1 score means low identification performance; therefore, the score means identification of these classes is difficult because of the similarity of the classes. Therefore, similarities in appearance can be calculated using Equation (7) and the *F*1 scores.

$$similarities\ between\ A\ and\ C = F1\ score\ between\ A\ and\ C = \frac{4ai + (a + 1)(c + g)}{(2a + c + g)(2i + c + g)} \tag{7}$$

Using Equation (7), we calculated the similarities among all classes and made a similarities matrix which shows the similarities in the value of each species, and applied

Ward's clustering method [36] to the similarities matrix and visualized the similarities relationships between classes using the dendrogram in R v.3.6.3 software.

## 3. Results

### 3.1. Validation 1: Performance for Dataset of Same Acquisition Date or Same Individual Trees

We first evaluated a model using a dataset which was obtained from the same acquisition date or same individual trees as the training dataset. Though this evaluation is rarely reproduced in practical applications, we revealed the potential for identifying each tree species. Therefore, the model showed a high validation accuracy: Kappa score was 0.979. The classification results are shown in Figure 6, and the details are presented in Supplementary Table S2. Most tree species showed over 90% accuracy in both precision and recall. This result means that when the data on light condition and trees are similar to those of the training data, most tree species can be identified. Meanwhile, classes 37, 41, and 50 showed a low classification accuracy. Class 41 had no recall and precision because there were no ground images. No ground images were obtained because of the program that separated each piece of training data into a validation dataset randomly at 30%, and a few overall images of the class.

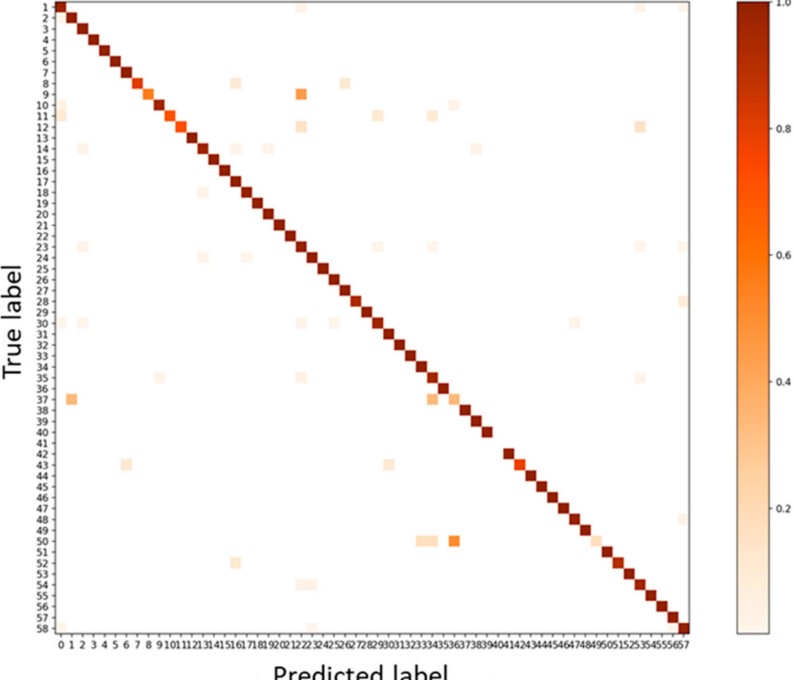

**Figure 6.** Confusion matrix of random separation dataset: same time or same trees. The vertical axis is the ground truth and the horizontal axis the model prediction. The number in each cell indicates the number of classified images; each cell is colored according to the percentage of the number of images in each class. The model showed 0.98 Kappa scores.

### 3.2. Validation 2: Performance for Dataset of Same Acquisition Date and Different Individual Trees

Next, we evaluated a model using a dataset which was obtained from the same acquisition date and different individual trees as the training dataset. From this evaluation, we revealed the spatial robustness of the model. Therefore, the model showed a relatively high accuracy, with a Kappa score of 0.715. This performance was approximately 25% lower than random separation. We show the details of the classification performance of each class (Figure 7, Supplementary Table S3). The precision and recall of 21 classes exceeded 50%. In particular, those exceeding 75% precision and recall were all classes of conifers (classes 1–5) and broadleaved *Castanopsis cuspidate*, *Quercus acuta*, *Acer carpinifolium*, *Acer mono Maxim* (classes 23, 26, 47, 48), and gap (class 58).

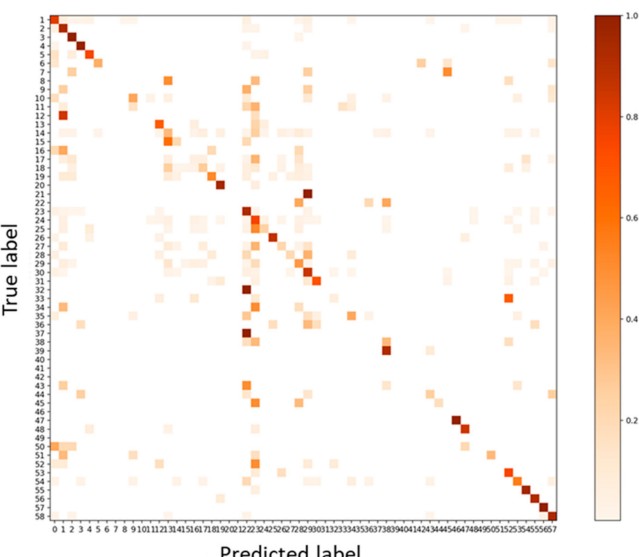

**Figure 7.** Confusion matrix of random separation dataset: same site and same time. The vertical axis is the ground truth and the horizontal axis is the model prediction. The number in each cell indicates the number of classified images; each cell is colored according to the percentage of the number of images in each class. The model yielded a Kappa score of 0.72.

### 3.3. Test: Performance for Dataset of Different Acquisition Date and Different Site

The model trained from different acquisition date and sites showed an even lower performance of a Kappa score of 0.472. We show the details of the classification performance of each class (Figure 8, Supplementary Table S4). Only three classes, *Chamaecyparis obtuse*, *Abies firma*, and gap (classes 1, 3, 58) showed a good performance, with both precision and recall exceeding 50%. Classes *Cryptomeria japonica*, *Pinus densiflora*, *Castanopsis cuspidate*, *Fagus crenata*, and *Quercus serrata* (classes 2, 4, 23, 24, 30) have the potential to be identified (precision or recall exceed 50%). However, it was found that the species representative of the stand type at each test site was at least 50% recall or precision.

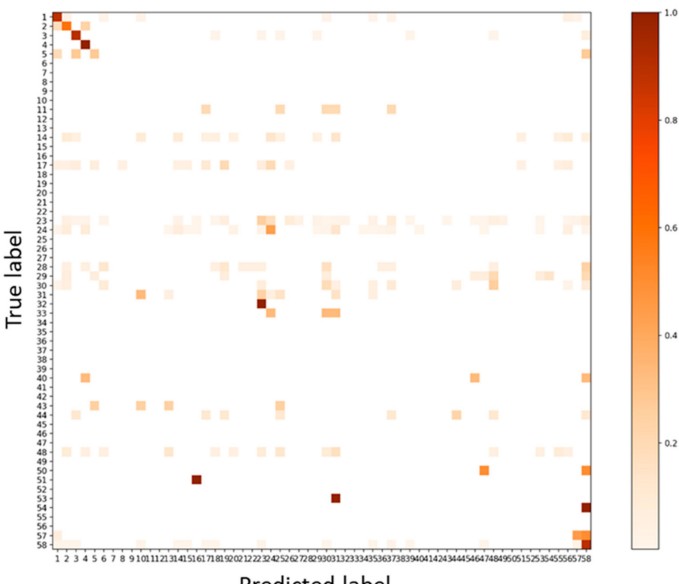

**Figure 8.** Classification result of the test. The vertical axis is the ground truth and the horizontal axis is the model prediction. The number in each cell indicates the number of classified images; each cell is colored according to the percentage of the number of images in each class. The Kappa score is 0.47.

### 3.4. Test Using Inventory Tuning

When we applied inventory tuning, the test result was improved from a Kappa score of 0.472 to 0.616. The classification details are presented in Figure 9 and Supplementary Table S5. Both precision and recall were over 50% in four out of five conifer species, *Chamaecyparis obtuse, Cryptomeria japonica, Abies firma* and *Pinus densiflora* (classes 1–4), broadleaved *Castanopsis cuspidate, Fagus crenata* and *Quercus serrata* (classes 23, 24, 30), and gap (class 58). These species representative of the stand type at each test site were over 50% in both recall and precision. Furthermore, *Tsuga sieboldii, Carpinus laxiflora*, and dead tree (classes 5, 17, 57) showed over 50% in recall or precision.

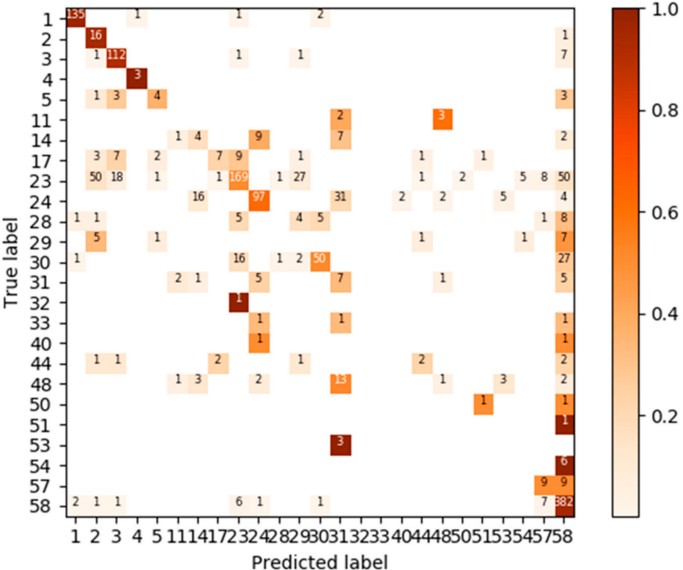

**Figure 9.** Classification result of the test using inventory tuning. The vertical axis is the ground truth and the horizontal axis is the model prediction. The number in each cell indicates the number of classified images; each cell is colored according to the percentage of the number of images in each class. The Kappa score is 0.62.

Finally, we visualized the classification results of CNN with inventory tuning applied to the Kasugayama site in GIS (Figure 10). We know that the Kasugayama, which is a primeval forest, is dominated by *Castanopsis cuspidate, Abies firma*, and *Cryptomeria japonica*.

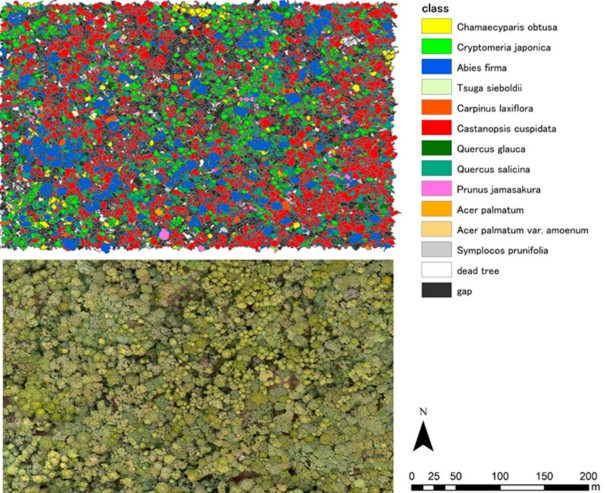

**Figure 10.** Orthomosaic photo and classification maps obtained with the inventory tuning CNN in the Kasugayama site. The CNN model was trained from a dataset of different sites. The area is 12 ha.

### 3.5. The Relationship between Accuracy and the Number of Training Images

The relationship between precision or recall accuracy and the number of training images is shown in Figure 11. Regarding validation 1, the plot showed a clear proportionality. In particular, classes that have less than 50 training samples sometimes show an accuracy ranging from 0.2 to 0.9, and classes with over 50 training samples always show high precision and recall of over 0.9. Regarding validation 2, we can confirm proportionality, but the plot was more scattered. The class that had fewer than 300 training images showed varied scores, but the class with over 300 training images always showed the potential for being identified, where both precision and recall were over 0.5. In addition, recall scores were higher than the precision in those classes. As for the test, most classes with less than 300 training samples showed zero precision and recall accuracy. The class with over 300 training samples always showed over 0.1 precision or recall. This means that at least 300 training samples should be gathered for practical use in this method.

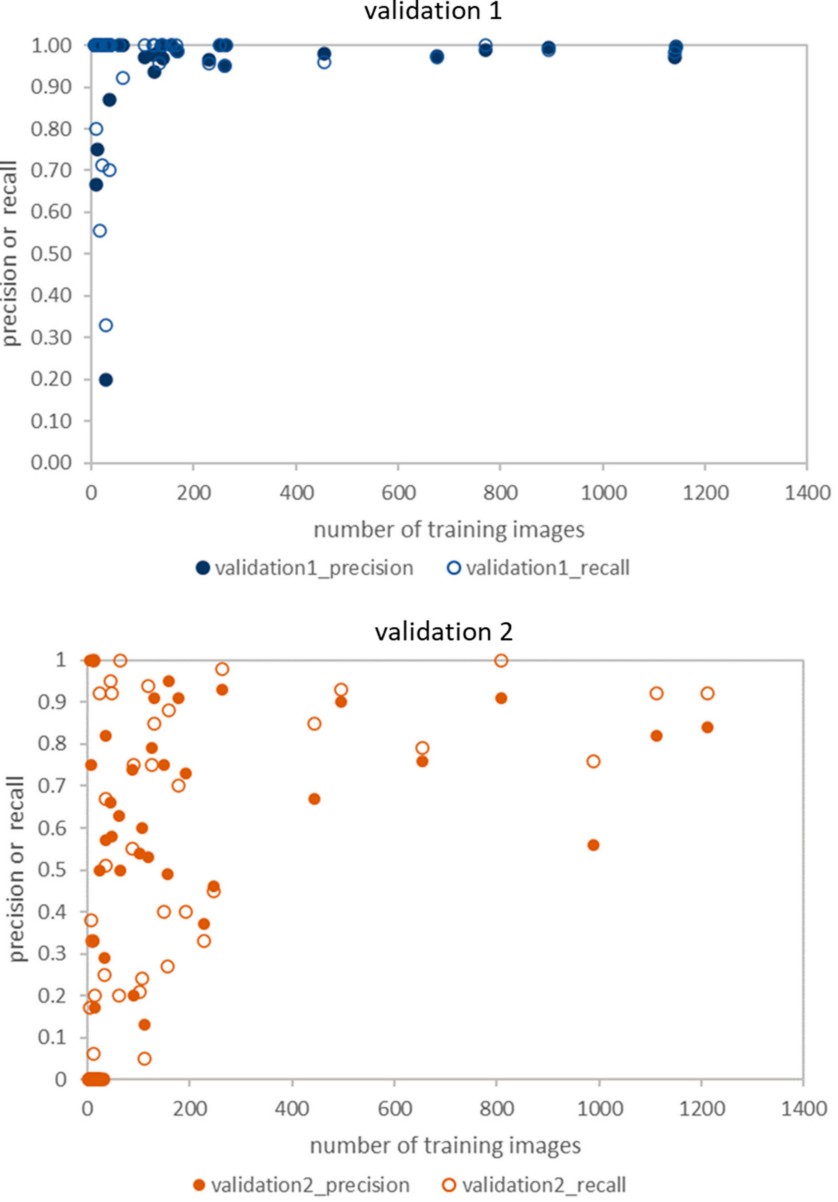

**Figure 11.** *Cont.*

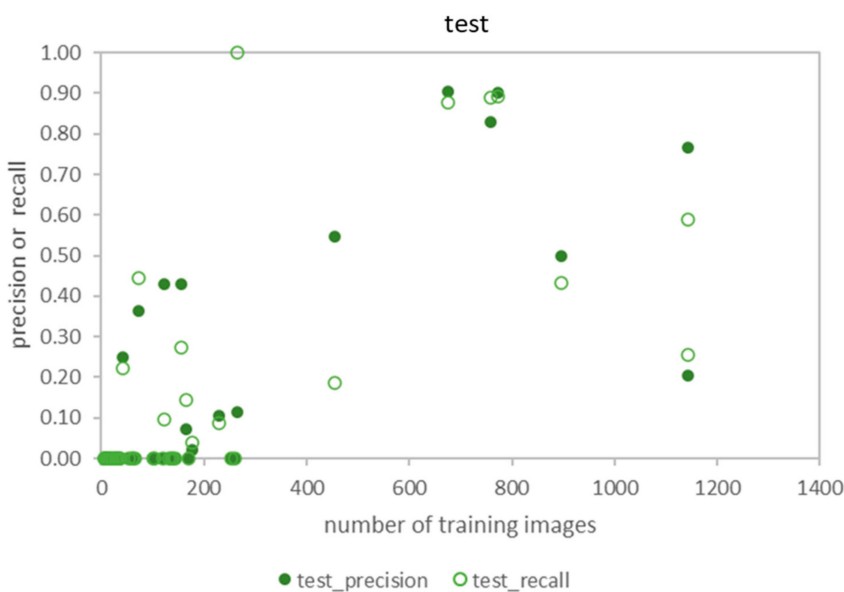

**Figure 11.** Relationship between precision or recall accuracy and number of training images. From top to bottom, the accuracy of validation 1, validation 2 and the test.

*3.6. Similarities in Appearance of Trees Species*

We visualized the similarities in appearance of each tree species from the classification results of the test in Figure 12. Although some species appear to be lined up chaotically, certain branches showed interesting tendencies. First, most branches on the trees in closest proximity to the center had large leaves (*Kalopanax septemlobus*, *Acer mono Maxim*, *Magnolia obovata*, *Pterocarya rhoifolia, and Aesculus turbinata*). *Kalopanax septemlobus* and *Acer mono Maxim* belong to different families, but both have large and palmate leaves. Therefore, the textures that can be seen from the UAV appear to be similar. The leaves of *Magnolia obovata*, *Pterocarya rhoifolia*, *Aesculus turbinata*, and the leaves of *Magnolia obovata* and *Aesculus turbinata* are similar to those of verticillate big leaves. However, *Pterocarya rhoifolia* has a pinnately compound leaf; therefore, *Pterocarya rhoifolia* is thought to be different, but in the images, the pinnately compound leaf looks similar to *Aesculus turbinata* at 100 m altitude. Both *Aesculus turbinata* and *Pterocarya rhoifolia* prefer a wet environment along the stream and grow at the same site. Therefore, on a macro level, the trees of the same functional types and those that thrive in the same environment may be hypothesized as having similar light acquisition performance and textures. In contrast, the phylogenetically close species were also located close to the dendrogram. For example, *Carpinus cordata* and *Carpinus laxiflora*, in proximity, and *Acer carpinifolium* and *Acer palmatum* are also located close in the dendrogram. Furthermore, *Chamaecyparis obtusa*, *Cryptomeria japonica*, and *Pinus densiflora* are coniferous trees positioned close to each other. Meanwhile, some coniferous trees are located close to the broadleaved trees in the dendrogram. This suggests that coniferous tree does not have unique textures from 100 m and above, even if the leaf shape differs from the broadleaved tree. Furthermore, some evergreen and deciduous trees were located in a disorderly manner, and some evergreen broadleaved trees were located close (such as *Ilex chinensis* and *Quercus glauca*, *Custanopsis cuspidate* and *Quercus acuta*). Therefore, although there may be common features among some evergreen trees, these are not enough for identification in summer.

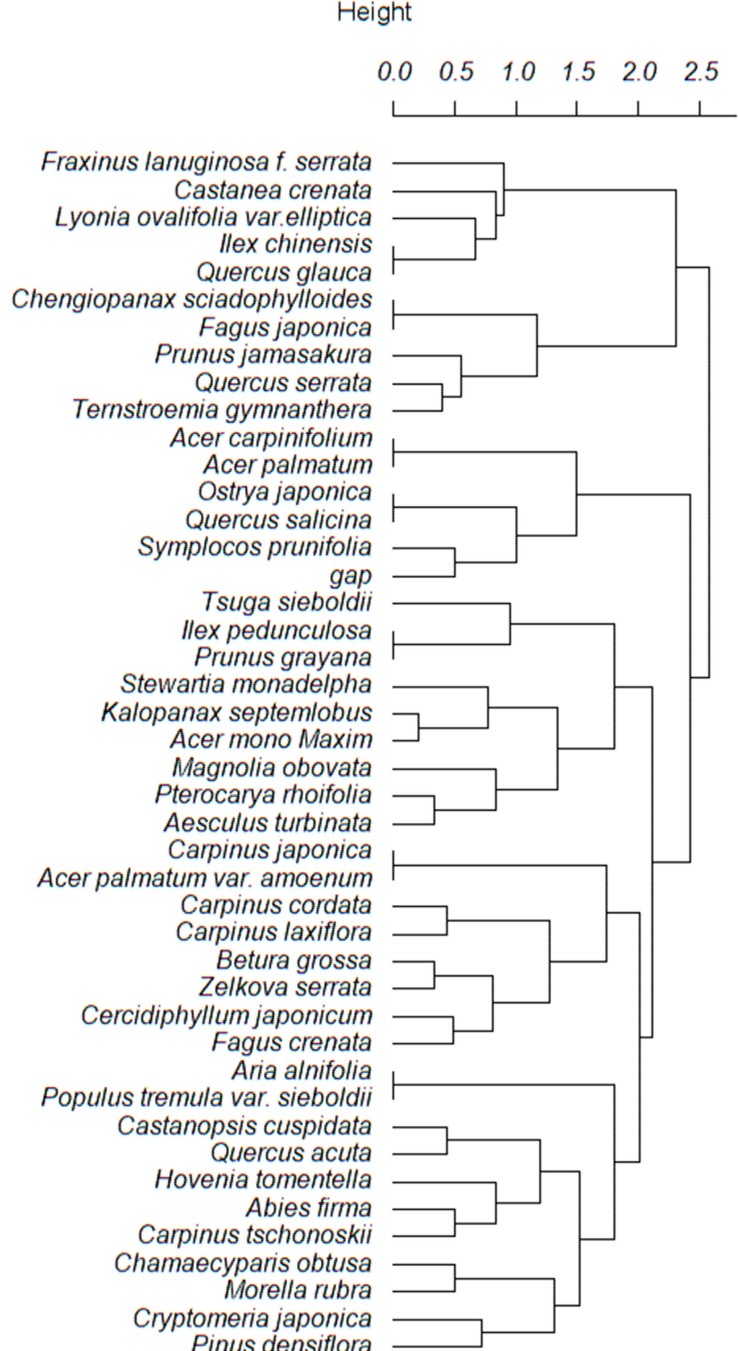

**Figure 12.** Similarities in appearance of trees species. This dendrogram was made from the distance calculated from *F*1 score of each class in test classification result.

## 4. Discussion

### 4.1. Classification Performance and Robustness

This study illustrates high potential and some stable robustness of the tree species identification systems using UAVs and deep learning. If the condition of the trees and shooting was optimal, most tree species could be identified (Kappa 0.979). Thus, this method could identify many tree species. However, this identification performance is rarely reproduced in practical applications. A similar classification accuracy is observed when this system is used for regular monitoring of the site.

When we used the dataset obtained simultaneously but from different trees, the performance decreased from a Kappa score of 0.979 to 0.715, and 36% (21 class) tree species

could be identified (over 50% in both precision and recall). This classification accuracy is assumed when we take images from one flight and separate them into training and test datasets. Though there are a few previous studies which evaluated the identification accuracy for over 20 species, Zhang et al. [15] showed the Kappa value of 0.66 for 40 tree classes and over 50% in both precision and recall for 38 classes, and Dalponte et al. [37] showed a 0.89 Kappa score for 22 tree classes and one shadow class using hyperspectral sensors and LiDAR. Our results bear comparison with those studies using Kappa scores, but 64% of classes did not show over 50% accuracy in both precision and recall, which is a lower performance than that seen in those studies. Except for the differences in sensors, there are differences in machine learning methods. In this study, we applied CNN, which can extract detailed features such as edge shape of foliage and bush of branches, hierarchical branching patterns, and outlines of tree shapes [29]. However, the method requires a large amount of training data, and for some classes where only small amount of training data was available, the classification performance was lower.

As for the test that shows the performance of general practical use, the accuracy was further decreased to a Kappa score of 0.472. Conversely, some tree species, especially coniferous trees and representative species of stands, showed potential for identification. This indicates that our system has temporal and spatial robustness to a certain extent. The identification from hyperspectral sensors can be affected by weather conditions such as shadows and illumination [5,6,15], and conditions of under vegetation, soil and litter, density of leaves, and health of trees [8,16–18]. For hyperspectral data analysis, building models which invariant to those differences is a grand challenge [38]. From this viewpoint, our method showed some robustness in some different conditions of weather and vegetation. Specifically illumination and growing environment, size of trees, and under vegetations can be thought to be different within dataset or sites, but for more precise discussion, we need detailed environmental and vegetation data. This robustness is likely preserved because using deep learning on high-resolution images allows identification based on tree features [29]. Therefore, in actual use, other researchers or general users could identify tree species automatically from this system. However, some conditions, such as wind strength, flight and camera parameters, and tree phenology, may affect the quality of the orthomosaic photo and classification accuracy. While these effects need to be evaluated, it is expected that deep learning may show some robustness to these effects if there are enough training data.

### 4.2. Inventory Tuning

In this study, we developed a new identification system called "inventory tuning". Using this method, we successfully created a local model from one large global model and improved the identification accuracy (0.472 to 0.616 Kappa score). The advantage of this method is that we can create a local model without retraining the initial model. Regarding the feasibility of the classification system, selecting the tree species and retraining the model is expected to be feasible, but it requires considerable computational load and time. From this perspective, inventory tuning does not require retraining and requires almost no computational load and time compared to the normal classification. Although in this study we used the field-obtained inventory data for inventory tuning, we may use the vegetation inventory data predicted from the longitude, latitude, and altitude for automatic inventory tuning in the future.

### 4.3. Similarities in Appearance of Trees Species

Based on the ease of error among the tree species, we developed a method to represent the similarities of the appearance of those tree species from a deep learning perspective in a dendrogram. From this method, we revealed that: (1) trees that belong to phylogenetically close species sometimes have similar textures, (2) tree species with similar leaf shapes sometimes look similar, and (3) tree species that prefer the same environment may sometimes show similar textures. Furthermore, tree types such as coniferous and broadleaved

or evergreen and deciduous do not always show common features among the tree type. A previous study using hyperspectral sensors sometimes faced challenges of identifying genetically close species in case (1) [39,40]. However, in particular, case (2) may be unique to this method, which uses UAV images and deep learning. Identification errors related to cases (2) and (3) can be reduced by increasing the resolution of the image. Furthermore, in this study, we used some fragmented tree images for training and test giving priority to automatic segmentation. Using only whole tree images will improve the coniferous tree species identification performance and clearly locate them far from a broadleaved tree in the dendrogram judging from the existence of tree tops and the difference in branching patterns. To distinguish deciduous trees from evergreen trees, autumn leaf coloring season is best, but detailed deciduous tree species classification may be difficult due to the unstable number of leaves and colors. Training and identification using multi-temporal data is also considered effective and is expected to be robust in terms of the impact of such phenology and accuracy.

### *4.4. Future Study*

Although we revealed the potential and robustness of identification system using UAVs and deep learning, there are technical challenges for forest management applications.

In this study, we segmented each tree crown automatically and applied deep learning to each crown image. However, our method failed to segment various tree crown perfectly, and it led to misclassification due to the fragmentation of some tree crown images. In the remote sensing field, the most common tree crown segmentation method combines local maximum filter and region-growing algorithms or watershed segmentation algorithms [41–46]. This method can be applied to height information of LiDAR or digital images from UAVs, especially in the coniferous forest [47,48]. In broadleaved forests or mixed forests, tree crown segmentation is still a difficult task because the size of trees has various range values and the tree tops are unclear; therefore, the development of the segmentation method, which does not need to set parameters such as pixel sizes, has been desired [49,50]. On this issue, a masking region method using deep learning such as instance segmentation [51] could help tree crown segmentation in broadleaved or mixed forests. This breakthrough will improve the performance of tree species identification from the air.

This system can be applied not only for forest areas but also urban areas. For applications in urban areas, we should apply some image processing for reducing computational load because there is a large amount of unnecessary images such as artificial objects. For the image processing, vegetation indices such as GRVI [52] and RGBVI [53] which can be calculated from RGB pixel value of images might be used for separating trees from objects.

## 5. Conclusions

This study demonstrated the stable robustness of the tree species identification system using UAV RGB imagery and deep learning for several tree species of temperate forest in Japan. As in previous studies, the discrimination performance was high using data obtained at the same acquisition timing, but in this study a model trained at other sites also successfully identified coniferous trees and species representative of the stands at different sites over 50% of recall or precision accuracy. Furthermore, it was found that using vegetation data in advance to limit the output of the model can improve accuracy from a Kappa score of 0.472 to 0.616 by allowing local models to be created simply and without retraining. For robust identification, some training data (300 samples in our method) is necessary, and if such data can be collected, the identification system will be accurate and widely usable. These findings will promote the practicalization of identification systems using UAV RGB images and deep learning. Furthermore, a tree species map made from the identification system will help to monitor native vegetation and invasive species, biodiversity assessment, and ecology research.

**Supplementary Materials:** The following supporting information can be downloaded at: https://www.mdpi.com/article/10.3390/rs14071710/s1, Table S1: Number of individual tree crown images of each dataset, which segmented from orthomosaic photo; Table S2: Details on classification accuracy of the model that was trained from the same time or same trees; Table S3: Details on classification accuracy of the model which trained from same time and different site; Table S4: Details on classification accuracy of the model which was trained from different times and different sites; Table S5: Details on classification accuracy of inventory tuning.

**Author Contributions:** M.O. conceived the whole project. M.O., S.W., T.N. and T.I. conducted acquisition of data. M.O. analyzed the data. All authors discussed the results and commented on the manuscript. All authors have read and agreed to the published version of the manuscript.

**Funding:** This research was funded by JSPS KAKENHI Grant Number JP19J22591 and JST PRESTO Grant Number JPMJPR15O1.

**Institutional Review Board Statement:** Not applicable.

**Informed Consent Statement:** Not applicable.

**Data Availability Statement:** The data that support the findings of this study are available from the corresponding author, M.O., upon reasonable request.

**Acknowledgments:** The authors would like to thank Hisashi Hasegawa who supported the research of Wakayama; Takeshi Torimaru and Naoyuki Nishimura with help with Daisen; Yusuke Onoda for supported the research of Higashiyama. We would also like to thank the staff of Kamigamo experimental forest, and Ashiu and Wakayama forest research station.

**Conflicts of Interest:** The authors declare no conflict of interest associated with this manuscript.

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
