# Peer review of "Practicality and Robustness of Tree Species Identification Using UAV RGB Image and Deep Learning in Temperate Forest in Japan"

_remotesensing, doi:10.3390/rs14071710_

Round 1

Reviewer 1 Report

This study used UAV to examine the identification of trees in the forests of Japan. The study is interesting, well organized, and well written and I believe it should be published in the journal Remote Sensing. However, there are few minor issues that should be resolved and they are as follows:

  • The authors should discuss use of LiDAR and Hyperspectral data for not only detecting tree properties in forests, but also in other settings such as urban areas. The following references should be included in such discussions:

           http://www.sciencedirect.com/science/article/pii/S161886671500076X

           https://www.mdpi.com/2072-4292/6/11/10636/pdf

  • The authors should discuss how and why the six sites were chosen for the study in Section 2.
  • In the discussion section, the authors should highlight how the results of this study compare to previous studies.
  • In the discussion section, I would like the authors to discuss the applicability of their method for trees in urban areas

Author Response

We deeply appreciate the time and effort you have put into reviewing our paper. All of your comments are highly valuable. As you will see below we have revised and improved the paper as a result of your valuable feedback. Changes to the manuscript can be confirmed with Red font. Please see the attachment.

Reviewer 2 Report

The manuscript presents the results of a classification study to detect species composition in several forest sites using a common UAV-RGB system and Deep Learning. Much has already been published in this research field, so that the readership can expect an advanced depth of content and methodological approach. The title suggests that the study is particularly concerned with the practical applicability and robustness of the method which is an interesting and important aspect. Further, it seems that the authors trained their algorithm on a large dataset on temperate tree species which is relatively rare. However, the paper has major shortcomings in its current form, making it difficult to extract the general hypothesis, specific research objectives and key messages. Major weaknesses can be found throughout all sections. Please have a close look at the following issues:

Introduction

- keep it broader from the beginning instead of rushing into different sensor systems (l. 34 f.) which have not been studied in this study (e.g., hyperspectral); rather start with: Why is tree species mapping important? What is the current status in Japan? --> write several sentences

- make sure not to mix up multispectral and hyperspectral systems; hyperspectral systems are still very rare

- l. 60 (whole paragraph): Does this paragraph refer to all of the above-mentioned studies? If yes, check if this is actually true as it represents a major depreciation of your cited literature.

- l. 75: this statement, which should be noted as hypothesis, requires further explanation; texture may be affected by windy conditions and/or flight and camera parameters (GSD, flight speed, shutter), e.g. motion blur; you have to elaborate on these aspects or at least mention that they are important to consider; also, the geometrical quality of the image is important which is usually hindered in forests due to lower GNSS accuracy

- Rewrite the specific objectives, l. 92, so that they are more precise; for example, what is meant by “time” – is it acquisition date, time of day or season or even similar lighting conditions?

Materials and methods

- It is unclear whether the sites have similar DBH, stems per ha, basal area – please add this information to Table 1; these attributes may also explain some of your results

- Provide insights into the georeferencing routine (section 2.2) – did you use GCPs? What was their accuracy? How many reference points?

- Provide further information on the field survey (section 2.3): How was the location of each tree assessed (device, accuracy)? How many trees have been surveyed? Maybe cite existing studies.

- Figure 2 is confusing to me: Why was there no georeferencing involved for the test data? What does “Tree A Images” and “Tree N images” mean? What does “Flight M” mean? What is the difference between “Orthomosaic photo” and “orthophoto”?

- Section 2.5 and Figure 4: The procedure needs further explanation. What does “same time or same trees” mean? Do you mean acquisition time and tree species? Or tree individuals? Please specify; What does “Tree A”, “Tree B” etc. mean? I suppose “Tree species A”, e.g. class 1.

- Table: 3: What are “images” – individual photos or orthophotos or segmented orthophotos (I suppose the latter)? - Please indicate in the text. Also, this table should be summarized in few sentences and shown in detail in the Appendix. You could also only show the details for the main species and the whole table in the Appendix.

- A method called “inventory tuning” is mentioned; it is unclear to me how this procedure works, please give further details. What was the “vegetation data of the target area”?

- The order of Sections 2.5 and 2.6 may be switched

Results

This section does not provide enough text to explain the results. In fact, the discussion section contains many paragraphs that should be put here; maybe focus on dominant species and provide full detail in the Appendix which may sharpen your results and discussion.

- l. 301: this sentence should be written in the conclusions

Discussion

This section is rather another results section than a scientific discussion. Discussing the results also means relating the results to other studies at a higher level of abstraction which I cannot see at the moment. Thus, the discussion should not rise new findings. I therefore recommend putting Figures 10-11 into the results section. You will then have to provide substantial new content to this section. You should for example discuss on the performance of the algorithm (comparison to other studies), quality of training dataset, uncertainties etc. You could also more precisely dicuss the “similarity” of species.

Conclusions

I cannot see a conclusion here and at the moment, this section is rather a summary or second abstract. What is the take-home message for the reader? What has to be achieved that the method reaches “practicalization”?

Language-related issues

Special focus should also be targeted to the current style of writing, which is often imprecise, rather subjective and non-scientific. Examples:

  1. 32: "from the air" --> remote sensing of tree species
  2. 54: "it costs millions of yen per scan" - imprecise and general statement- you could just say they are expensive and say the same; also, yen will not be familiar currency for all readers - think of an international readership
  3. 69: "general digital cameras" - imprecise, define general
  4. 74: "which is a famous commercial UAV" - use scientific language please, rather say "frequently used"
  5. 299: "Finally, I visualised…" – please avoid first person singular in scientific writing (also, there are several authors written on the paper)
  6. 447: Avoid the terms “stable robustness” and “stable performance” as they do not contain quantitative information; e.g. rather state your kappa scores.

Author Response

We deeply appreciate your careful reading and effort. We think our paper became better because all of your comments are highly professional and valuable. Thanks again for your advice that will help us in our future research. As you will see below we have revised and improved the paper as a result of your beneficial feedback. Changes to the manuscript can be confirmed with Red font.

Round 2

Reviewer 2 Report

The authors have addressed my comments in a target-oriented and conscientious manner. The scientific quality of the paper has improved considerably - congratulations!

I have only a few minor suggestions for changes:

l. 407: "images of same the individual tree" --> images of the same individual tree
l. 415: "dataset was shown" --> dataset is shown
l. 808: the sentence seems incomplete. I suggest to write two sentences here. Suggestion: "In this study, we applied CNN which can extract detailed features such as edge shape of foliage and bush of branches, hierarchical branching patterns, and outlines of tree shapes [29]. However, the method requires a large amount of training data and for some classes with only small amount of training data available, the classification performance was lower."
l. 851: please cite the "previous study"
l. 865: Though --> Although

Author Response

We deeply appreciate your careful reading and effort. Please see the attachment.
